# To Explore the Stem Cells Homing to GBM: The Rise to the Occasion

**DOI:** 10.3390/biomedicines10050986

**Published:** 2022-04-24

**Authors:** Sergey Tsibulnikov, Natalya M. Drefs, Peter S. Timashev, Ilya V. Ulasov

**Affiliations:** 1Group of Experimental Biotherapy and Diagnostic, Institute for Regenerative Medicine, Sechenov First Moscow State Medical University, Trubetskaya 8-2, 119991 Moscow, Russia; ser-tsibulnikov@yandex.ru; 2TS Oncology, 125047 Moscow, Russia; natalia.drefs@tsoncology.com; 3World-Class Research Center “Digital Biodesign and Personalized Healthcare”, Sechenov First Moscow State Medical University, Trubetskaya 8-2, 119991 Moscow, Russia; timashev_p_s@staff.sechenov.ru; 4Department of Advanced Materials, Institute for Regenerative Medicine, Sechenov First Moscow State Medical University, 119991 Moscow, Russia

**Keywords:** stem cells, glioblastoma, CXCR4

## Abstract

Multiple efforts are currently underway to develop targeted therapeutic deliveries to the site of glioblastoma progression. The use of carriers represents advancement in the delivery of various therapeutic agents as a new approach in neuro-oncology. Mesenchymal stem cells (MSCs) and neural stem cells (NSCs) are used because of their capability in migrating and delivering therapeutic payloads to tumors. Two of the main properties that carrier cells should possess are their ability to specifically migrate from the bloodstream and low immunogenicity. In this article, we also compared the morphological and molecular features of each type of stem cell that underlie their migration capacity to glioblastoma. Thus, the major focus of the current review is on proteins and lipid molecules that are released by GBM to attract stem cells.

## 1. Introduction

Glioblastoma (GBM) is the most common and aggressive primary tumor in adults. According to the classification of the World Health Organization (WHO), glioblastoma belongs to the fourth type of malignancy of intracranial neoplasms. Even though this type of tumor is the most common representative of primary brain tumors, the incidence of the disease in humans is relatively low—3.1 cases per 100,000 population (for comparison, 171.2 and 201.4 cases per 100,000 population were noted for breast and prostate tumors, respectively) [1]. It is well-established that the uncontrolled growth of glial cells stimulates aggressive tumor growth that may lead to the death of patients. Anticancer therapy could increase the survival of patients with neoplasms. A group of scientists demonstrated that the survival rate within five years does not exceed 5%, and the median survival rate reaches 15 months after treatment, including surgery combined with radiation and chemotherapy. As shown earlier [2], the low survival rate of patients with GBM is based on the adaptive response of tumor cells to therapy. An adaptive response is expressed in the activation of the growth of tumor cells by an autocrine mechanism and the induction of cellular factors that cause inflammation and immunosuppression of the tumor [3,4,5]. The standard methods of treatment do not increase the survival rate. Experimental methods of glioblastoma therapy are being actively developed that reach the effectiveness of the existing ones. On the other hand [6], the use of cancer vaccines, which are based on the activation of the acquired immunity of a patient in response to tumor-specific antigens [7], is promising for experimental oncology. A vaccine can be both individual peptides [8,9], which is promising for experimental oncology, and whole autologous antigen-presenting cells [10]. However, the use of a platform based on human and animal viruses seems to be a more effective approach to inducing an immune response in tumors. This is explained by the fact that the virus can be used to deliver and express a library of tumor antigens and succinates the activation of the immune response and the elimination of tumors and metastases. On the other hand, the natural mechanism of viral replication makes it possible to obtain recombinant proteins in significant volumes. There are approaches involving the use of viruses, both as vectors for gene therapy [11] and as oncolytic agents [12], delivering immunogenic proteins or antisense RNAs to tumor proteins that promote carcinogenesis [13]. At the same time, native immune responses can be one of the obstacles that decrease the anticancer efficacy of viral vectors against cancer cells [6]. For example, the pool of the neutralization of antibodies against human adenovirus type 5 decreases the efficacy of CRAd (Conditionally replicated adenoviruses)-based vectors using the human adenovirus type 5 genome [14,15]. Additionally, low diffusion through the tumor mass [16] and the activation of cellular protein kinases [17], for example, modulate the inflammatory immune response and overall decrease the therapeutic response to the selected vectors. To overcome those issues and still specifically deliver a gene therapy vector to the target cells, stem cell carriers can be of interest.

It is known that delivering proteins to pathological areas minimizes the possible off-target effects of drugs and highlights the advantages of such carriers for anti-GBM therapy [18,19]. Therefore, the design of approaches that seek GBM tumors and lack a strong immune response [20,21] can be a promising anticancer advancement. Consequently, using such promising candidates as mesenchymal and neural stem cells, demonstrating tumor chemotaxis and the ability to deliver therapeutic payloads to glioblastoma, seems convenient.

## 2. General Characteristics of Mesenchymal Stem Cells (MSC)

Mesenchymal stem cells (MSCs) are a heterogeneous population of fibroblast-like cells. MSCs represent a small population (0.01% of all nucleated cells found in bone marrow) [22]. MSCs are multipotent and differentiate into chondrocytes, skeletal myocytes, and neurons, if applicable. As noted earlier, more than 95% of MSCs express a high level of CD73, CD90, and CD105 and a low level of protein II (the major histocompatibility complex (MHC-II)) and CD45, CD34, CD14, CD11b, and CD19 on their surface. MSCs can express other markers in various combinations dependent on the tissue specificity. In 2011, Ode et al. provided a mechanistic insight into the biological significance of surface receptors for MSC migration. CD73/CD29 were among the antigens that were described in the study and implemented in MSC migration. It turns out that MSC migration is controlled by CD73/CD29 and their downstream targets such as Lck, Fyn, and Src-family kinase activation [23].

## 3. MSC—Cell Vectors for Migration and Delivery of Therapeutic Proteins

MSCs expressing protein molecules have been used in preclinical studies to successfully deliver antigens and induce a therapeutic effect in human and animal glioblastoma cells. As previously stated, stem cells were used to deliver viruses [24,25,26], suicide proteins from herpes simplex viruses like thymidine kinase (TK) [27,28,29], cellular miRNA [30], exosomes [31], apoptosis-induced proteins like TNF (tumor necrosis factor)-related apoptosis-inducing ligand (TRAIL) [32], cytosine deaminase (CD) [33], carboxylesterase 2 (sCE2, a prodrug-activating enzyme) [34], antiangiogenic thrombospondin-1 [35], and chemical agents such as paclitaxel [36] that activates the T-cell response in glioblastomas [37] (Figure 1).

Typically, MSCs are effective in delivering therapeutic proteins into diffuse tumors such as glioblastoma. This fact improves the following activation of the migration and adhesion of genetically modified MSC due to the concentration gradient mediated by SDF-1α, TGF-β1 (transforming growth factor beta), and CCL2 (C-C motif ligand 2) around the tumor. MSC demonstrated an increased migratory potential towards tumor cells [38,39,40]. Thus, MSC migration depends mainly on the expression of the CCR2 and CXCR4 receptors, since their inhibition with specific antibodies leads to a decrease in orchestrated cellular movements [41,42]. At the molecular level, binding ligands to these receptors enhances the expression of transcription factors in MSCs, which is required for stem cell differentiation [43,44]. Furthermore, Pavon et al. [42] demonstrated the endpoint of MSC migration to GBM, which is the area enriched with CD133-positive stem cells (GSC) (Figure 1). It has been shown by multiple investigators that GSCs responsible for tumor growth in vivo [45,46,47] and resistance to therapy [48] are located in anatomical structures. In such a periarteriolar niche [49] with elevated SDF-1α, various cathepsin [50] expressions and, adjacent to the arteriolar tunica adventitia, the CXCR4-enrcihed GSC interact with various cells, including MSC [51] (Figure 2). It has also been proposed that CXCR4, which is highly expressed in MSCs and endothelial cells (EC) [52], upregulates the expression of BDNF and EOMES genes, both of which are important for GSC stemness [52], inhibits the antitumor immune response caused by T-cell exhaustion, and regulates IFN-γ [53] expression. Although, in the glioma niche, the scientific evidence points to MSC-mediated direct interactions with GSC via the paracrine way [54], the presence of immune cells, astrocytes, and/or neural stem/precursor cells (NPC) in GBM [55] might also help to maintain the GSC phenotype and aggressiveness. Considering the localization of GSC in the perialveolar niche surrounded by hypoxic and highly vascularized regions, the targeting and eradication of GSC represent a clinical challenge.

## 4. Methods and Factors That Increase the Migration Properties of MSCs

Earlier reports suggest that in vitro cultivation conditions can influence the cell surface receptor level [62]. Therefore, the relationship between the types of MSCs of different origins and their ability to migrate into A549 tumors [63], ovarian [64], and colon [65] cancer cells is manifestative. Recently, it has been shown that MSCs have enhanced the tropism towards glioblastoma cells treated with ionizing radiation [66]. This therapy increases the secretion of chemokines and cytokines above the basal level [67]. Even though tumor cells induce SDF-1α [42,62,63] and CCL2, the existing gradient of chemokines and cytokines provides the migration of a small group of MSCs. The findings from earlier research showed that, when MSCs are systemically injected into laboratory animals with intracerebral glioblastoma, less than 1% of the MSCs migrate to GBM [68,69,70], leaving open questions about the fate of most injected MSCs at the injection/implantation site and the possibility of enhancing the migration of all MSCs into the tumor area.

## 5. Modeling MSC Migration In Vitro

MSC exposure to tissue growth factor-beta (TGF-β) [62] or challenging MSCs with monocyte chemotactic protein-1 (MCP-1), chemokine (C-C motif) ligand 8 (CCL-8), and/or interleukin 8 (IL-8) [71] can increase the migration of MSCs towards the tumor. Another way that increases the migration of MSCs into the tumor is the ability of MSCs to secrete metalloproteinases to cleave the basement membranes of cells and penetrate deep into the tumor. It was experimentally confirmed that cell incubation in a medium containing complement component 1q (C1q) [72], a mixture of erythropoietin and granulocyte colony-stimulating factor (G-CSF) [73], positively affects the expression of metalloproteinases. In this case, the simultaneous addition of erythropoietin and GM-CSF (1 IU/mL and 0.1 ng/mL, respectively) to the medium increased the MMP-2 (matrix metalloproteinase-2) expression, while the number of other metalloproteinases, as well as metalloproteinase inhibitors, remained unchanged. The ability of cells to migrate was also measured if a factor was added to the medium. Cells that were exposed to the medium with the addition of both factors at the same concentration showed the greatest ability to migrate and increase the metalloproteinase expression. At this concentration of substances, the concentration of extracellular signal-regulated kinase (ERK1/2) proteins significantly increased as well. These conditions indicate a possible effect of the ERK1/2 signaling pathway on the migration abilities of MSCs.

The short-term exposure of cells to valproic acid increases the expression of CXCR4, C1q, and MMP2 and improves the migration towards SDF-1 [74]. Additionally, it was shown that the exposure of MSCs to C1q increases the migration activity towards SDF-1 (20 ng/mL)-level migration to SDF-1 at a higher concentration (100 ng/mL). At the same time, CXCR4 expression on the cell surface increased from 1.5% to 9.5%. The study showed that short-term (3 or 6 h) exposure of MSCs to valproic acid at a concentration of 10 mmol increases the expression of CXCR4 mRNA by 40 and 60 times, respectively. The amount of CXCR4 was not increased on the cell surface, which confirmed the results of another study. In this study, the majority of CXCR4 was isolated intracellularly and formed heterodimers with CXCR7 [75]. Cells exposed to valproic acid migrated towards low (20 ng/mL) and high (100 ng/mL) SDF-1α gradients two and four times higher, respectively. When using the AMD3100-CXCR4 antagonist, the tropism for SDF-1α almost completely disappeared. The statement that an increased tropism due to cell exposure to valproate is associated with CXCR4 was confirmed. To increase the CXCR4 receptor on the cell surface, insulin-like growth factor 1 (IGF-1); tumor necrosis factor-alpha (TNF-α); and interleukin 1 beta (IL-1β), interferon-gamma (IFN-γ), or glycogen synthase-3 beta (glycogen synthase kinase 3 beta, GSK-3β) were added. It has been shown that the addition of complement 1q to the medium increases cell migration in response to the SDF-1 concentration gradients [72].

## 6. Induced Hypoxia

Hypoxia is the main feature of the neoplastic process caused by glioblastoma, as well as immunosuppression and inflammation. The induction of hypoxia is based on the molecular interactions of tumor microcirculation with tumor cells. As a result of this process, GBM cells gain phenotypic properties that allow them to survive under stress and become resistant [76]. GBM cells are known for an extensive angiogenesis, mostly mediated by HIF-1α. Besides their roles in neovascularization, HIF-1α and HIF-2α initiate the induction of a panel of stemness genes [77], chemokines, and their receptors, including CXCR4, TGF-β, CCL2, and CXCL12 [41,43]. It was noted, in some studies, that the induction of CXCR4 and CXCR7 mRNA was observed in MSC cells in the case of hypoxia (3% oxygen) due to an increase in the expression of hypoxia-induced factor (HIF-1α). The experiment was carried out to study the direct effect of hypoxic conditions on the migration properties of MSCs. The cells were first cultivated under normal conditions or hypoxic conditions, and then, the cells in containers with various concentrations (1–100 ng/mL) of SDF-1α were placed for 6 h for analysis in a Boyden chamber. As a result, hypoxia increased the migration of the studied cells at any concentration of the chemokine. Additionally, studies have shown that hypoxia enhances the adhesive properties of MSCs (by an average of 2.55 times). Further, MSCs were cultured under normal conditions or under hypoxic conditions in combination with antibodies to CXCR4 and CXCR7 to determine the role of CXCR4 and CXCR7 in cell migration, adhesion, and survival. Blocking at least one of the receptors reduces cell adhesion, inhibits CXCR4 and the migration properties of MSCs, and in the case of CXCR7, decreases cell survival. Thus, it was demonstrated that the CXCR4 receptor plays a leading role in cell migration in the case of hypoxia. Its activity is regulated by HIF-1α. The results confirmed the assumption that an overexpression of CXCR4 and CXCR7 occurs through HIF-1α activation and affects the migration and adhesion properties, as well as the survival of MSCs. Another signaling pathway that regulates the activity and expression of CXCR4 and CXCR7 is PI3K/AKT. The PI3K/AKT inhibitor allowed researchers to examine CXCR4 and CXCR7 expression in relation to HIF-1α. It turned out that the inhibition of PI3K/AKT leads to a decrease in the expression of HIF-1α, which, in turn, affects the expression of CXCR4 and CXCR7 [41]. Considering that HIF-1α also regulates the expression of MMP 2, MT1-MMP [78] enhances the adipogenic and osteogenic differentiation in some types of MSCs [79]. It can be concluded that it has a complex effect on the activity of the intracellular signaling pathways responsible for MSC adhesion and migration.

## 7. MSCs Have Been Genetically Modified to Increase Their Migration and Survival in Tumors

MSC migration is considered an important factor for the experimental treatment of GBM, as it increases survival in the tumor mass under hypoxic conditions. However, some scientists have noted that MSC transplantation does not cause long-term survival inside the tumor. Thus, it is necessary to find solutions that can productively increase the effectiveness of cell therapy. One such approach was proposed by Hu et al. [80]. In that study, the authors proposed to create a stable MSC clone expressing miR-211 upon lentivirus infection. At the molecular level, this miRNA regulated the activity of the STAT signaling pathway and, in particular, the stability of STAT5A mRNA. Thus, during in vivo experiments, it was possible to increase the survival rate of MSCs in heart muscle damage due to miR-211 expression. Similar experiments were performed by using MSCs for the experimental treatment of GBM with miR-4731 [81]. Furthermore, strategies for increasing MSC migration to GBM were proposed by cloning CXCR4 [82], an integrin that increases adhesion, into the genome [83].

Due to their special properties, MSCs are one of the main candidates for delivery vehicles for therapeutic proteins. One of these properties is the ability of MSCs to taxi for tumors [84,85] and high immunocompatibility [86], which makes allogeneic transplantation possible. Additionally, an undoubted advantage is an ease of obtaining these cells due to the relatively low invasiveness and the absence of ethical problems; they can be obtained from bone marrow, adipose tissue, cord blood, umbilical cord, and placental tissue [87]. Now, data from various sources indicate the relative safety of administering MSCs to the human body [86]. The risk of side effects (embolism and allergic reactions) can be minimized [88] by following the recommendations during the route of administration and cell concentrations. Many factors may influence potential tumorigenesis after MSC transplantation, including donor age, host tissue, growth regulators expressed in the recipient tissue, and mechanisms that control MSC behavior at the target site. In addition, the manipulation and long-term cultivation of MSCs in vitro can cause genetic instability and chromosomal aberrations. The response against many cumulative factors can be spontaneous tumor transformation. Patients who have transplanted stem cells often receive long-term chemotherapy or radiation therapy, so their immune systems do not work properly, which may lead to tumor development in the body.

## 8. General Characteristics of Neural Stem Cells (NSCs)

Neural stem cells are undifferentiated nerve cells that are defined based on their extensive replicative potential, their ability to differentiate into different types of neurons and glial cells of the central nervous system, and their ability to sustain long-term self-renewal [89]. In the adult mammalian CNS, active neurogenesis occurs in two separate regions: the subgranular zone (SGZ) of the dentate gyrus in the hippocampus of granule cells and the subventricular zone (SVZ) of the lateral ventricles in the forebrain of interneurons in the olfactory bulb [90]. Additionally, it is suggested that there is another zone in which NSC niches can be located—the hypothalamus [91]. It has been established that the proximity of the tumor to the SVZ directly affects the survival of patients [92], which may indicate the effect of NSC on tumor development, such as GBM [93]. Cells from the SVZ have highly expressed CXCR4 and SDF-1 markers, which have been used to preserve cell migration [94], differentiation [95], proliferation [94], and stemness [96]. Mechanistically, Liao et al. [97], for the first time, showed that CXCL12 controls the mesenchymal activation of cell signaling that is relevant to the Snail, N-cadherin, and NF-kB pathways. Most recently, Goffart et al. [98] demonstrated that CXCL12 controls mesenchymal activation via the upregulation of numerous transcriptional factors and markers, such as ESR1, ZEB2, SOX10, FOXC2, CDH2, FN1, and vimentin. Of interest, Jin et al. [99] showed that circRNA EPHB4 modulates glioma stemness via the upregulation of SOX10 and miR637 interactions.

As previously mentioned, the rapid growth of tumor cells, high concentrations of chemokines, and cytokines secreted by the tumor microenvironment and tumor cells are the most common causes of glioblastoma progression. Reportedly, the induction of a whole group of chemokines controls the migration of healthy cells into the tumor, coupled with immunosuppression and the formation of niches for glioblastoma cells. According to the literature, SDF-1 and its CXCR4 receptor are found in the niches of glioblastoma stem cells [100]. Anatomically, this niche also serves for the maturation of neural progenitor cells, so two groups of cells use the same chemokine signals for their transformation [101]. Thus, stem cell migration due to SDF-1α gradients and the suppression of tumor growth by delivering an antitumor protein will reduce the progression of GMB in tumor niches. Various types of stem cells have been used to treat GBM because of their ability to migrate, deliver recombinant antitumor proteins, and secrete antitumor-enhancing factors. 

## 9. NSC—Cell Vectors for Migration into the Tumor

NSC has a tropism for tumors [102]. Thus, recent investigations have compared the NSC characteristics of migration to glioma tumor cells in comparison with MSC and showed that the differences in the number of migrated cells are statistically insignificant [103]. In comparison with NSCs, the MSCs do not develop a rolling stage of transmigration from the bloodstream (Figure 3). This fact is explained by the absence of fucose in the composition of glycoproteins expressed on the cell surface [104]. As a result, glycoproteins do not interact with selectins on the surfaces of endothelial cells, despite the artificial fucosylation of glycoproteins, leading to a decrease in cell migration [105]. Moreover, after this modification, the number of dead cells increases. One study showed that NSCs cannot tolerate shear stress, unlike MSCs and hematopoietic stem cells (HSCs). The fucosylation of HSCs improved the migration and did not affect the viability, which explained the worsened viability of NSCs. Diaz-Coranguez et al. [104] studied the transmigration of NSCs to tumor cells in vitro. It was shown that NSC transmigration increased in the model with tumor cells. There was no rolling stage in this model. The value of the transendothelial electrical resistance (TEER) decreased compared to the control, in which the cells migrated to astrocytes. Additionally, holes were formed on the borders between endothelial cells but were absent from the control. It has been shown that NSCs migrate through these holes.

## 10. Methods and Factors That Increase the Migration Properties of NSC

MCP1 [106] is a chemoattractant for NSCs and other extracellular matrix proteins, including tenascin, laminin, fibronectin, vitronectin, and various types of collagens [107]. These extracellular matrix components are intensively produced by the glioblastoma stroma in the areas where tumors develop [108,109]. Laminin and tenascin are the most important for NSC migration [107]. Together, these proteins take part in the regulation of cell-to-cell signaling, cell adhesion, migration, and mechanical resilience [110]. At the molecular level, secreted extracellular matrix proteins control the maintenance and longevity of stem cells through TAG1 [111], Notch [112], and cadherins [113,114], suggesting a regulatory role in tissue homeostasis. Numerous ascending pathways in cells regulate NSC activity through transcription factors or cross-factor signaling interactions. Experimental culturing demonstrated the sensitivity of NSCs to the action of VEGF. It is known that the basal level of VEGF in the sample with glioma was 7.6 times higher than in the control cells, which explains the decrease in transmigration when we use antibodies that neutralize the VEGF [104]. At the same time, inhibition of the VEGF led to the re-establishing of TEER values in GBM cells at the initial level, which means the participation of VEGF in the regulation of cell-to-cell contacts. Early studies on the GBM model showed a lack of relationship between the level of EGF and the efficiency of NSC transmigration, although the TEER value of the endothelium increased if the medium contained EGF with an unchanged expression of claudin 5 [104].

Epidemiological and experimental data showed clearly that glioblastoma cells produce biologically active substances that provide progression (such as MMPs) and resistance [115,116]. Most tumor cells from primary samples demonstrate a pronounced increase in cyclohexinase [115] and prostaglandin 2E [116], which stimulates chronic inflammation and tumor resistance. Researchers in recent studies measured the amounts of prostaglandin E2 in healthy brain cells (astrocytes) and tumor cells of glioblastoma. It turned out that prostaglandin E2 reduces the value of TEER and inhibits the increase of this potential in EGF [104]. Astrocyte samples had a 17% more prostaglandin increase compared to glioma samples. However, given the almost complete absence of EGF in the medium with glioma, this amount of prostaglandin can significantly affect the TEER value. On the other hand, the use of a COX-2 inhibitor also led to a decrease in the amount of prostaglandin and an increase in the TEER value. It should be noted that the levels of the IFN-α, TNF-α, IL-12p70, IL-1β, IL-6, IL-8, and IL-10 cytokines in the glioma and astrocytes samples did not differ, which disproved the involvement of interleukins in NSC migration.

## 11. Conclusions

Due to the lack of many experiments directly comparing the migration properties of MSCs and NSCs, it is rather difficult to unambiguously identify the most suitable candidate for the transfer of therapeutic agents to the tumor because of the lack of such experiments. To address that issue, we summarized the known information from clinical trials using NSC and MSC cells (Table 1), a list of factors expressed in glioblastoma, and a general scheme of the therapeutical agents that can be delivered by NSC or MSC cells to glioma stem cells (Figure 2). Thus, studies have shown that NSCs do not respond well to tangential stress [105]. As a result, migration from the bloodstream is difficult. Furthermore, given the possible link between NSCs in the SVZ zone and GBM progression in several nearby anatomical areas [92], special caution should be exercised in the study design to avoid facilitating tumor progression and NSC transformation [117]. Although various studies have pointed out the anti-glioma therapeutic effect of the designed NSCs [118], some investigators, such as Wang et al. [119], showed that NSCs promote the proliferation, migration, invasion, and tumorigenicity of glioblastoma cells in vitro and in vivo. Although it is known that a 3-month-old embryo was the source for NSCs and isolated cells were able to produce neurospheres and exhibited a positive expression of SOX2, CD133, and NESTIN and were negative for GFAP, NeuN, and CD68 markers, there is still limited information on whether these cells meet some criteria for NSCs to demonstrate an anticancer effect. From the scientific literature, it is known that the expression of EGFR [117], selection of the most efficient time for GBM treatment [120], status of reprogramming with a viral vector that contributes to gene integration into the cellular genome [121], etc. can be critical for the anticancer effect mediated by NSCs. On the other hand, long culture NSCs [122] undergo aging, which is characterized by an upregulation of proinflammatory markers and a transient or stable induction of transcriptional factors that contribute to reprogramming and therapy resistance [123]. In the case of MSCs, these cells are adapted for transmigration [124] through the endothelium to tumors [125]. We must therefore be cautious to avoid MSC reprogramming by tumor cells, which eventually contribute to the tumor growth and spread. Several studies have reported that MSC can have both tumorigenic and antitumorigenic properties. As an anti-glioma effect of MSC results in inhibiting cell growth, promoting glioma differentiation [126] to increase tumor cell sensitivity to TMZ induces cell apoptosis [127] and cycle arrest. MSC can increase the invasiveness (due to hyaluronic acid deposition [128] or MMP activity [129]) and migration [130], as well as promote cell proliferation [128] and stemness [131]. The data by Bajetto et al. showed that MSC interactions (cocultures with glioma cells or releasing soluble factors) might produce a divergent effect from the inhibition of tumor growth to the strong production of molecules involved in inflammation, angiogenesis, cell migration, and proliferation, such as IL-8, GRO, ENA-78, and IL-6 [132]. Therefore, the safety issues prior to using MSC or NSC as an anticancer approach require special attention, since putting a stop to cell reprogramming mediated by tumor cells and brain environment might prevent reversing the phenotype and extending the therapeutic options. One method of enabling the control of the administrated cells would be to introduce suicide genes (for example, TK or CD [133]) into the cell genome, which can be activated to prevent a possible risk from stem cells for the long term. In that case, the development of a microfluidic system will assess the contribution to migration of the transmigration stage through blood vessels and will allow cell-type comparisons. Further research in the field of the cell-based delivery of therapeutic agents should be focused on improving the migration properties of cell carriers (Table 2), as well as an in-depth investigation of their interactions with tumor cells, in order to minimize the negative effects of the therapy.

## Figures and Tables

**Figure 1 biomedicines-10-00986-f001:**
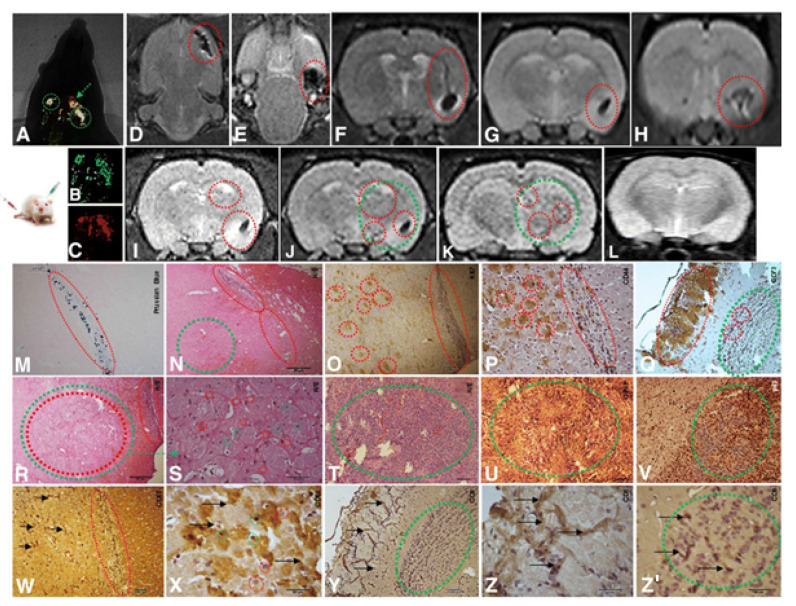
Implantation of 1 × 104 CD133+ GBM cells labeled Qdots (705 nm). After the establishment of the GBM (28 days), there was an infusion in caudal vein 1 × 104 MSCs (MION-Rh); the development of the tumor was followed for 20 days. (**A**) MSCs labeled MION-Rh and CD133+ GBM cells labeled Qdot 705 nm using combined fluorescence and X-ray detection. (**B**) CD133+ GBM cells labeled Qdot 705 nm and visualized by fluorescence detection. (**C**) MSCs labeled MION-Rh and visualized by fluorescence detection. (**D**–**L**) MRI (T2*-weighted images) of animal brain monitoring of the process of migration of MSCs, which were able to cross the blood–brain barrier of the animal and migrated to the tumor region, promoting GBM cell proliferation. (**L**) MRI (T2*-weighted images) of the animal brain without the stereotaxic implantation of cells (control group). The red dot circle shows migration assays of MSCs, and the green dot circle shows tumor propagation. (**M**) IHC analysis for Prussian blue staining of the MSCs labeled with MION-Rh. (**N**,**R**–**T**) Hematoxylin and eosin staining. (**U**) IHC analysis for GFAP. (**O**) IHC analysis for Ki67. (**V**) IHC analysis for p53. (**P**) IHC analysis for CD44 staining of the MSCs. (**Q**) IHC analysis for CD73 staining of the MSCs. (**W**,**X**) IHC analysis for CD63 staining of the MSC-derived exosomes. (**Y**,**Z**,**Z**′) IHC analysis for CD9 staining of the MSCs-derived exosomes. Scale, 40 µm. These images are representative of all the collected MSCs and GBM samples. Image and text were copied from the original study by Pavon et al. [42] and distributed under Attribution 4.0 International (CC BY 4.0).

**Figure 2 biomedicines-10-00986-f002:**
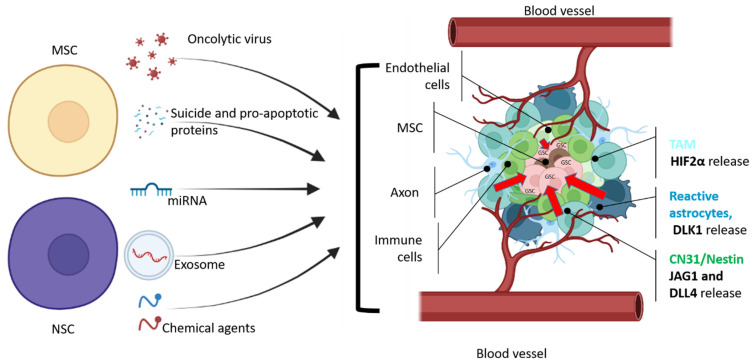
General scheme of therapeutic agents that can be delivered by NSC or MSC cells to glioma stem cells. GSCs are located in the necrotic area of the tumor, which is surrounded by hypoxic and vascularized regions with leaky blood vessels (adapted from Vinogradov et al. [56]). CD31^+^/Nestin^+^ and reactive astrocytes produce JAG1, DLL4 [57], and DLK1 [58] to perinecrotic and perivascular tumor regions to maintain GSC stemness and proliferation. According to Kvisten et al., [59] tumor-associated astrocyte (TMA) areas are rare in necrotic areas and more distributed in perivascular and perinecrotic areas of the GBM. Despite the morphological and phenotypical differences, TMA produces HIF-2α [60,61], which also controls the GBM pathobiology. In order to deliver a therapeutic payload, NSCs or MSCs should overcome multiple obstacles that prevent GSCs from being eliminated. Cell microenvironment releases growth factors to maintain GSC (red arrows).

**Figure 3 biomedicines-10-00986-f003:**
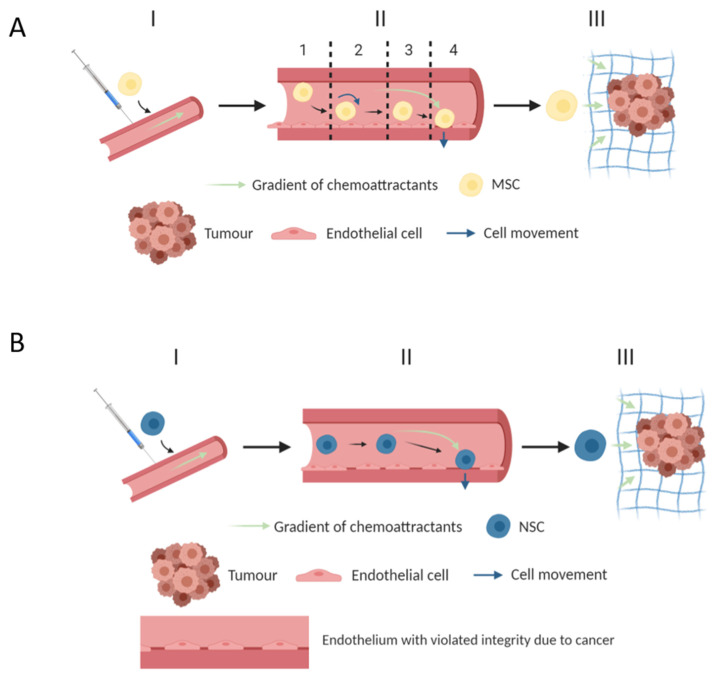
The migration of MSCs (**A**) and NSCs (**B**). (I) Chemotaxis-induced stem cell injection and migration. (II) The transmigration of stem cells from blood vessels to the tumor site. (III) Gaptotaxis and tumor internalization and penetration. 1—Rolling, 2—Cells activation, 3—Integrin-dependent uptake, and 4—Transmigration through the endothelium and basal membrane.

**Table 1 biomedicines-10-00986-t001:** List of the current preclinical and clinical trials used in the application of MSC and NSC against glioblastoma.

NSC				
Reference	Type of Study	Cargo	Treatment	Results
Chen et al., 2013	Clinical trial	-	Radiation therapy for SVZ	Radiation increases patient’s PFS and OS PFS:15.1 vs. 10.3 OS:17.5 vs. 15.6 (months)
Lee et al., 2013	Clinical trial	-	Radiation therapy for SVZ	Radiation improves PFS PFS:12.6 vs. 9.9
Portnow et al., 2017	Clinical trial	CD	HB1.F3.CD.C21; CD-NSCs + oral administration of 5-FC	Migration of NSC to the tumor and locally produces chemotherapy is confirmed.
Aboody et al., 2013	Preclinical study	CD	HB1.F3.CD.C21; CD-NSCs + intraperitoneal injection of 5-FC	Inhibition of GBM progression and prolongation of mice with GBM xenografts survival
Bago et al., 2016	Preclinical study	TRAIL	iNSC modified with TRAIL	Inhibition of GBM progression and prolongation of mice with GBM xenografts survival
Dey et al., 2016	Preclinical study	Oncolytic Adenovirus CRAd-S-pK7	Overexpressed CXCR4 in NSCs and loaded with CRAd-S-pk7	Increased mice survival with GBM xenografts
MSC				
NCT03896568	Clinical trial	Oncolytic Adenovirus DNX-2401	Mesenchymal stem cells loaded with a tumor selective oncolytic adenovirus, DNX-2401	Recruiting
NCT04657315	Clinical trial	CD	Mesenchymal stem cells into which cytosine deaminase the suicide gene was injected	Recruiting
Oraee-Yazdani et al., 2021	Clinical trial	HSV-TK	Autologous mesenchymal stem cells as HSV-TK gene vehicle	Biosafety of MSC PFS:23.7 OS: 32.2 (months)
Mohme et al., 2020	Preclinical study	IL-12, IL-7	Intra-tumoral of genetically modified MSCs that co-express high levels of IL-12 and IL-7	Intra-tumoral administration of MSC IL7/12 induced significant tumor growth inhibition and remission of established intracranial tumors
Novak et al., 2020	Preclinical study	TRAIL	TRAIL-secreting MSC/nanomedicine spheroid system	The hybrid spheroid inhibited the tumor growth efficiently
Shimizu et al., 2021	Preclinical study	Oncolytic Adenovirus Delta-24-RGD	Bone marrow-derived human mesenchymal stem cells loaded with Delta-24-RGD	Intravascular administration of PD-BM-MSC-D24 increased the survival of mice harboring U87MG gliomas

Abbreviations: MSC—Mesenchymal stem cells, DNX2401—former Delta-24-RGD oncolytic self-replicated adenoviral vector, BM—Bone marrow-derived, PD-1—program cell death protein 1, TRAIL—TNF-related apoptosis-inducing ligand, PFS—progression-free survival, OS—overall survival, HSV—herpes simplex virus, NSC—Neural stem cells, TK—thymidine kinase, CXCR4—C-X-C chemokine receptor type 4, and CRAd—Conditionally replicated adenoviruses.

**Table 2 biomedicines-10-00986-t002:** Growth factors expressed in glioblastoma and their roles in therapy and patient survival.

Factors	Receptor Presence at the Stem Cells		Clinical Significance for the Patients with GBM	
Tumor-Derived Factors	MSC	NSC	Impact for Therapy/References	Correlation with Survival
SDF-1	+ [134]	+ [135,136,137]	Inhibition of SDF-1α enhances anti-VEGF therapy [138]	
IL6	+ [134]		Inhibition Il6 negatively affects GBM viability	No correlation [139]
IL1β	+ [134]			
HGF	+ [84]	+ [137,140,141]	Inhibits tumor stem-like cells [142]	
VEGF	+ [141]	+ [136,140]	Anti-VEGF treatment inhibits angiogenesis and strongly increases cell invasion and tumor hypoxia [143]	[144]
uPA		+ [136]	Inhibition uPA/uPAR attenuates invasion, angiogenesis in glioblastoma cells [145]	[146]
PDGFAA/BB	+ [66,147]	+ [148]	Ablation of PDGF signaling sensitizes anti-VEGF/VEGFR treatment in GBM [149]	[139]
FGF ligands		+ [148]	FGF1/FGFR signaling axis sustains the stem cell characteristics of GBM cells [150]	
EGF	+ [151]	+ [140,148]	Inhibition EGF suppresses GBM migration [152]	
IGF	+ [147]	+ [148]	Inhibition IGF reduces GBM proliferation [153]	[139]
Annexin A2		+	Annexin A2 acts at multiple levels in determining the disseminating and aggressive behavior of GBM cells [154]	[154]
TNF-a	+ [134]	+	EGFR plus TNF inhibition is effective in TMZ-resistant recurrent GBM [155]	No correlation [139]
TGF b	+ [134]	+ [148]	TGF-β1 modulates temozolomide resistance in glioblastoma [156]	[157] No correlation [139]

Abbreviations: “+”-presence on the surface of the cells, SDF-1—Stromal cell-derived factor-1, HGF—Hepatocyte growth factor, VEGF—Vascular endothelial growth factor, uPA—Urokinase, PDGF—Platelet-derived growth factor, FGF-1—acidic fibroblast growth factor, TNF—Tumor necrosis factor, and TGF b—Transforming growth factor beta.

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
