# Peer review of "To Explore the Stem Cells Homing to GBM: The Rise to the Occasion"

_biomedicines, 2022, doi:10.3390/biomedicines10050986_

Round 1

Reviewer 1 Report

This is a well-written and interesting review manuscript. However, the functions and dynamics of MSCs are not properly covered and described. Therefore, more details should be included in a revised version of the manuscript. In addition, clear figures are needed to explain the interactions between MSCs and glioblastoma (stem) cells.

- Morphologically, where are mesencymal stem cells (MSCs) localized in glioblastoma tumors? Please elaborate on this in more detail. Include microscopic images of MSCs in glioblastoma tumors.

- Glioblastoma stem cells (GSCs) and GSC niches should be described in more detail, especially because GSCs express CXCR4 in SDF-1α-rich niches. For more informatie see:

o Hira VVV, Wormer JR, Kakar H, et al. Periarteriolar Glioblastoma Stem Cell Niches Express Bone Marrow Hematopoietic Stem Cell Niche Proteins. Journal of Histochemistry & Cytochemistry. 2018;66(3):155-173.

o Hira VVV, Breznik B, Vittori M, Loncq de Jong A, Mlakar J, Oostra RJ, Khurshed M, Molenaar RJ, Lah T, Van Noorden CJF. Similarities Between Stem Cell Niches in Glioblastoma and Bone Marrow: Rays of Hope for Novel Treatment Strategies. J Histochem Cytochem. 2020;68(1):33-57.

- The authors discuss MSCs in glioblastoma tumors, but MSCs are not mentioned in relation to GSC niches in glioblastoma tumors. GSCs interact with MSCs in peri-arteriolar GSC niches in GBM tumors. For more info see Hira et al 2020. MSCs express high levels of intracellular DF-1α and are attracted to CXCR4-positive GSCs. For more info see Hira et al 2018 and Hira et al, 2020.

o Hira VVV, Wormer JR, Kakar H, et al. Periarteriolar Glioblastoma Stem Cell Niches Express Bone Marrow Hematopoietic Stem Cell Niche Proteins. Journal of Histochemistry & Cytochemistry. 2018;66(3):155-173.

o Hira VVV, Breznik B, Vittori M, Loncq de Jong A, Mlakar J, Oostra RJ, Khurshed M, Molenaar RJ, Lah T, Van Noorden CJF. Similarities Between Stem Cell Niches in Glioblastoma and Bone Marrow: Rays of Hope for Novel Treatment Strategies. J Histochem Cytochem. 2020;68(1):33-57.

- What other cell types do MSCs interact with in glioblastoma tumors?

- Are MSCs involved in glioblastoma (stem) cell resistance via SDF-1α-CXCR4 interactions?

- A clear schematic illustration of interactions between MSCs/NSCs and glioblastoma cells is missing. Please add a clear figure and include the effects of hypoxia on the cell-cell interactions.

- Do the authors consider MSCs as a potential vehicle for drug delivery in glioblastoma tumors? MSCs have been reported to be pro-tumorigenic as well as anti-tumerigenic. Please elaborate on this in the Discussion section and if appropriate include these references.

o Barcellos-de-Souza P, Gori V, Bambi F, Chiarugi P. Tumor microenvironment: bone marrow-mesenchymal stem cells as key players. Biochim Biophys Acta. 2013; 1836:321-35.

o Klopp AH, Gupta A, Spaeth E, Andreeff M, Marini F. Dissecting a discrepancy in the iterature: do mesenchymal stem cells support or suppress tumor growth? Stem Cells. 2011; 29:11-9.

- MSCs can activate matrix metalloproteinases (MMPs), resulting in more cellular invasion of cancer cells. Please elaborate on the risks of using MSCs as therapeutic drug delivery tool.

- Line 116: Which cancer cell lines were used in these experiments?

- Line 211: Include details on the neural stem cells (NSCs) and their niches (the subventricular zone and subgranular zone). Describe how CXCR4 and SDF-1α are involved in the maintenance of NSC stemness in the SVZ.

- Line 226: Neural stem cells (NSCs) demonstrate tropism to glioblastoma tumors. What are the effects of infiltrating NSCs on glioblastoma (stem) cell proliferation/migration/invasion/survival? Please elaborate on this in more detail.

Author Response

EXPERT RESPONSE

Expert-1:

It is a well-written and interesting review manuscript. However, the functions and dynamics of MSCs are not properly covered and described. Therefore, more details should be included in a revised version of the manuscript. In addition, clear figures are needed to explain the interactions between MSCs and glioblastoma (stem) cells.

Our response: We would like to thank the reviewer for his/her evaluation of our work. We will try our best to enrich the manuscript as per reviewer’s suggestion.

Concern No.-1: Morphologically, where are mesenchymal stem cells (MSCs) localized in glioblastoma tumors? Please elaborate on this in more detail. Include microscopic images of MSCs in glioblastoma tumors.

Our response: We thank the reviewer for this suggestion. We have added few more information as evidence that MSC and GSC are localized in GSC niches (Lines:107-109, Fig. 1.).

Concern No.-2: Glioblastoma stem cells (GSCs) and GSC niches should be described in more detail, especially because GSCs express CXCR4 in SDF-1α-rich niches. For more information see:

o Hira VVV, Wormer JR, Kakar H, et al. Periarteriolar Glioblastoma Stem Cell Niches Express Bone Marrow Hematopoietic Stem Cell Niche Proteins. Journal of Histochemistry & Cytochemistry. 2018;66(3):155-173.

o Hira VVV, Breznik B, Vittori M, Loncq de Jong A, Mlakar J, Oostra RJ, Khurshed M, Molenaar RJ, Lah T, Van Noorden CJF. Similarities Between Stem Cell Niches in Glioblastoma and Bone Marrow: Rays of Hope for Novel Treatment Strategies. J HistochemCytochem. 2020;68(1):33-57.

Our response: We appreciate the expert’s advice. As suggested, we provided in depth data that is related to GSC niches (Lines:107-118).

Concern No.-3: The authors discuss MSCs in glioblastoma tumors, but MSCs are not mentioned in relation to GSC niches in glioblastoma tumors. GSCs interact with MSCs in peri-arteriolar GSC niches in GBM tumors. For more info see Hira et al 2020. MSCs express high levels of intracellular DF-1α and are attracted to CXCR4-positive GSCs. For more info, see Hira et al 2018 and Hira et al, 2020.

Our response: We thank the reviewer for this thoughtful comment. In the revised manuscript (Lines: 108-112) we have added information as per reviewer’s suggestion.

Concern No.-4: What other cell types do MSCs interact with in glioblastoma tumors?

Our response: We value this suggestion by the reviewer and added information accordingly which suggest that MSC interact with glioma via paracrine mechanism in glioma niche, and no evidences of such interaction is presented so far regarding endothelial cells (EC), immune cells, astrocytes; and/or neural stem/precursor cells (NPC). We have addressed that point in our revised manuscript (Lines: 113-116).

Concern No.-5: Are MSCs involved in glioblastoma (stem) cell resistance via SDF-1α-CXCR4 interactions?

Our response: Authors thank reviewer for this comment. To address it, we added new information to the revised manuscript. CXCR4, which is highly, expressed in MSCs, upregulates the expression of BDNF and EOMES genes vital for GSC stemness (Lines: 110 - 112).

Concern No.-6: A clear schematic illustration of interactions between MSCs/NSCs and glioblastoma cells is missing. Please add a clear figure and include the effects of hypoxia on the cell-cell interactions.

Our response: In the revised manuscript, we have provided a new figure (Fig. 2). Authors believe that this enrichment in the manuscript may address the reviewer’s concern.

Concern No.-7: Do the authors consider MSCs as a potential vehicle for drug delivery in glioblastoma tumors? MSCs have been reported to be pro-tumorigenic as well as anti-tumerogenic. Please elaborate on this in the Discussion section and if appropriate include these references.

Our response: This aspect has been updated in the manuscript (Lines: 364-394). We also considered other investigators and their study to elaborate the role of MSC and NSC in glioma therapy and progression. Here are some references in this regard;

  • Barcellos-de-Souza P, Gori V, Bambi F, Chiarugi P. Tumor microenvironment: bone marrow-mesenchymal stem cells as key players. BiochimBiophys Acta. 2013; 1836:321-35.
  • Klopp AH, Gupta A, Spaeth E, Andreeff M, Marini F. Dissecting a discrepancy in the iterature: do mesenchymal stem cells support or suppress tumor growth? Stem Cells. 2011; 29:11-9.

Concern No.-8: MSCs can activate matrix metalloproteinases (MMPs), resulting in more cellular invasion of cancer cells. Please elaborate on the risks of using MSCs as therapeutic drug delivery tool.

Our response: Additional explanations regarding risks have been added at the end (Line: 387). Authors expect that reviewer may be convinced with this line.

Concern No.-9: Line 116 (old version): Which cancer cell lines were used in these experiments?

Our response: We have corrected this part of the text in the current revision (New Lines:141-142).

Concern No.-10: Line 211 (old version): Include details on the neural stem cells (NSCs) and their niches (the subventricular zone and subgranular zone). Describe how CXCR4 and SDF-1α are involved in the maintenance of NSC stemness in the SVZ.

Our response: We thank reviewer for the clarification. We have further described NSCs and their niches, and also mentioned the effect of CXCR4 on stem preservation (Lines: 264 -282).

Concern No.-11: Line 226 (old version): Neural stem cells (NSCs) demonstrate tropism to glioblastoma tumors. What are the effects of infiltrating NSCs on glioblastoma (stem) cell proliferation/migration/invasion/survival? Please elaborate on this in more detail.

Our response: Authors have tried to update this aspect of NSCs. Line (364-380)

Reviewer 2 Report

In the manuscript entitled “To explore the stem cells homing to GBM: the rise to the occasion” the authors aim to review the literature regarding the use of mesenchymal stem cells (MSC) and neuronal stem cells (NSC) as vectors for the delivery of new therapeutic agent against glioblastoma (GBM). The authors reported a superficial description of both stem cells, then, they try to describe some mechanism directing the homing of both types of stem cells to glioblastoma bulk. Despite the topic could be of great interest, the manuscript seems superficial trying to report some evidence without deepening a specific topic. The authors must revise the organization of the manuscript. The authors should divide the manuscript into two parts: one on MSC and one on NSC. Both these chapters should be divided into sub-chapters in which a specific mechanism involved in stem cell homing (e.g. CXCR4, SDF-1, TNF-a, hypoxia, miRNA…) will be investigated. Furthermore, there are several specific comments:

  • The abstract should be revised in terms of the English language and should be better focused on the different aspects reviewed in the manuscript
  • The authors stated the pivotal use of stem cells as delivery agents. The authors should use a paragraph to report in detail the types of cargo used in pre-clinical and clinical studies on glioblastoma.
  • Chapter 7, the authors only reported a sentence on the genetic modification used to improve MSC homing (line 192-193), without any detail on the method used and the obtained results.
  • The authors should improve the manuscript by introducing almost a figure and a table, which summarize the pre-clinical and clinical application of MSC and NSC cells.
  • The authors should improve the conclusion with the limitations and advantages of the MSC and NSC use. The authors should highlight the future directions to overcome all the reported limitations.

Author Response

EXPERT RESPONSE

Expert 2:

In the manuscript entitled “To explore the stem cells homing to GBM: the rise to the occasion” the authors aim to review the literature regarding the use of mesenchymal stem cells (MSC) and neuronal stem cells (NSC) as vectors for the delivery of new therapeutic agent against glioblastoma (GBM). The authors reported a superficial description of both stem cells, then, they try to describe some mechanism directing the homing of both types of stem cells to glioblastoma bulk. Despite the topic could be of great interest, the manuscript seems superficial trying to report some evidence without deepening a specific topic. The authors must revise the organization of the manuscript. The authors should divide the manuscript into two parts: one on MSC and one on NSC. Both these chapters should be divided into sub-chapters in which a specific mechanism involved in stem cell homing (e.g. CXCR4, SDF-1, TNF-a, hypoxia, miRNA…) will be investigated. Furthermore, there are several specific comments:

Concern No.-12: The abstract should be revised in terms of the English language and should be better focused on the different aspects reviewed in the manuscript.

Our response: We agreed with expert opinion and made our best effort to address the specifics of our topic.

Concern No.-13: The authors stated the pivotal use of stem cells as delivery agents. The authors should use a paragraph to report in detail the types of cargo used in pre-clinical and clinical studies on glioblastoma.

Our response: We summarized the preclinical and clinical studies using NSC or NSC (Table 1). In that table, we incorporated a column describing cargo for each of the stem cell studies.

Concern No.-14: Chapter 7, the authors only reported a sentence on the genetic modification used to improve MSC homing (Lines: 192-193, old version), without any detail on the method used and the obtained results.

Our response: We have incorporated some additional information into Line: 241.

Concern No.-15:The authors should improve the manuscript by introducing almost a figure and a table, which summarize the pre-clinical and clinical application of MSC and NSC cells.

Our response: Authors thank the reviewer for this valuable suggestion. In response to that comment, we incorporated the requested Table-1.

Concern No. 16: The authors should improve the conclusion with the limitations and advantages of the MSC and NSC use. The authors should highlight the future directions to overcome all the reported limitations.

Ourresponse: Based on your valuable suggestions, we have corrected the conclusion. Lines: 364 -394

Reviewer 3 Report

In the present article “To explore the stem cells homing to GBM: the rise to the occasion”, the authors explore the morphological and molecular features of each type of stem cell that underlie their migration capacity to glioblastoma along with special focus on protein and lipid molecules which released my GBM to attract stem cells. However, it is my opinion that this article in its present form is not yet ready for publication. For the favor of improving the paper, some points are listed below:

  1. Page 2. line number 50 to 54 authors mentioned that ..the use of a platform based on human and animal viruses seems to be a more effective approach to inducing an immune response further this is explained by the fact that the virus can be used to deliver and express a library of tumor antigens and succinates the activation of the immune response and elimination of tumors and metastases. In my openinon authors should also need to describe few points on the discrepancy of use of  antigens inducing factors that influnce the activation of the immune responses and how stem cells therapy is more efficacious.
  2. Page 2. line number 61 to 63…The ability to detect various signals, actively penetrate and deliver proteins in the pathological area, minimizing the off-target effects of drugs, highlight the advantages of using such carriers in the therapy of glioblastoma. These sentences are not clear, need to rephrase. Moreover, to make article more interesting, I would like to suggest authors to add figures that demonstrating the delivery of proteins to target the tumor and related molecular mechanism.
  3. Page 2. line number 74 to 76 authors mentioned that ..As noted earlier, more than 95% of MSCs express a high level of CD73, CD90, CD105, and a low level of protein II (the major histocompatibility complex  (MHC-II)) and SD45, SD34, SD14, SD11b, SD19 on their surface. how these markers play specific role in migration dependent on tissue specificity that need to elaborate more.
  4. Page 2. Last paragraph the sentence is not clear, it seems floating specifically line 90 to 94.  
  5. Full name of abbreviation is missing. Authors should mention the full name of evry abbreviation if they are using first time in text, please check these abbreviations in whole manuscript carefully.
  6. Page 2. Last paragraph, last sentence, authors mentioned that MSC migration depends mainly on the expression of CCR2 and CXCR4 receptors on the surface of migrating cells.. This line seems to confusing, in previous Paraph, line number 87 to 90 it is already mentioned that EGF/EGFRvIII[32] and CXCR4/SDF1alpha complexes help in stem cell They why authors calming .. MSC migration depends mainly on the expression of CCR2 and CXCR4 receptors.
  7. Add one table related to Medical conditions and summary of the known effects of factors -inducing cell migaration in therapies. 

Author Response

EXPERT RESPONSE

Expert 3:

In the present article “To explore the stem cells homing to GBM: the rise to the occasion”, the authors explore the morphological and molecular features of each type of stem cell that underlie their migration capacity to glioblastoma along with special focus on protein and lipid molecules which released my GBM to attract stem cells. However, it is my opinion that this article in its present form is not yet ready for publication. For the favor of improving the paper, some points are listed below:

Concern No.-17: Page 2. line number 50 to 54 (Old version) authors mentioned that the use of a platform based on human and animal viruses seems to be a more effective approach to inducing an immune response further this is explained by the fact that the virus can be used to deliver and express a library of tumor antigens and succinates the activation of the immune response and elimination of tumors and metastases. In my opinion authors should also need to describe few points on the discrepancy of use ofantigens inducing factors that influence the activation of the immune responses and how stem cells therapy is more efficacious.&

Our response: We took your suggestion into account and added the required information accordingly (Lines:56-66).

Concern No.-18: Page 2. line number 61 to 63( old version)…The ability to detect various signals, actively penetrate and deliver proteins in the pathological area, minimizing the off-target effects of drugs, highlight the advantages of using such carriers in the therapy of glioblastoma. These sentences are not clear, need to rephrase. Moreover, to make article more interesting, I would like to suggest authors to add figures that demonstrating the delivery of proteins to target the tumor and related molecular mechanism.&

Our response: As advised, we clarified our phrase at Lines: 67-71 of the revised manuscript.

In terms of the figure, we believe that the requested cartoon will be the best fit for the new general gene therapy review, which is now in preparation in our laboratory.

Concern No.-19: Page 2. line number 74 to 76 old version)authors mentioned that. As noted earlier, more than 95% of MSCs express a high level of on their surface. how these markers play specific role in migration dependent on tissue specificity that need to elaborate more.

Our response: We found that only CD73 is involved in the migration process through Lck and Fyn, Src-family kinases (Lines: 81 -84).

Concern No.-20:Page 2. Last paragraph the sentence is not clear; it seems floating specifically line 90 to 94 (old version).

Our response: We thank reviewer for this observation. We revised that text. Please review the current version presented at Lines: 67-72.

Concern No.21: Full name of abbreviation is missing. Authors should mention the full name of every abbreviation if they are using first time in text, please check these abbreviations in whole manuscript carefully.

Our response: We apologize for this flaw and have corrected abbreviations throughout the text.

Concern No.-22: Page 2. Last paragraph, last sentence, authors mentioned that MSC migration depends mainly on the expression of CCR2 and CXCR4 receptors on the surface of migrating cells. This line seems to confusing, in previous Paraph, line number 87 to 90 (old version) it is already mentioned that EGF/EGFRvIII [32] and CXCR4/SDF1alpha complexes help in stem cell They why authors calming ..MSC migration depends mainly on the expression of CCR2 and CXCR4 receptors.

Ourresponse: The text has been corrected as per suggestion by the reviewer. We have explained the position according to which these receptors are considered by us to be the most important in the context of the migratory properties of MSCs (Lines:94 -101).

Concern No.-23: Add one table related to Medical conditions and summary of the known effects of factors -inducing cell migration in therapies. 

Our response: Authors thank reviewer for such incorporation of information in the manuscript. We have accordingly added the requested table (Table 2).

Round 2

Reviewer 1 Report

The manuscript was improved and can now be published.

Author Response

Thank you for this opinion.IU

Reviewer 2 Report

The authors responded to all the raised concerns

Author Response

Thank you for all your hard work. IU